# Microbiological and Molecular Features Associated with Persistent and Relapsing *Staphylococcus aureus* Prosthetic Joint Infection

**DOI:** 10.3390/antibiotics11081119

**Published:** 2022-08-18

**Authors:** Irene Muñoz-Gallego, María Ángeles Meléndez-Carmona, Jaime Lora-Tamayo, Carlos Garrido-Allepuz, Fernando Chaves, Virginia Sebastián, Esther Viedma

**Affiliations:** 1Department of Clinical Microbiology, Hospital Universitario 12 de Octubre, Instituto de Investigación Biomédica “i + 12” Hospital 12 de Octubre, 28041 Madrid, Spain; 2Department of Internal Medicine, Hospital Universitario 12 de Octubre, Instituto de Investigación Biomédica “i + 12” Hospital 12 de Octubre, 28041 Madrid, Spain; 3CIBER de Enfermedades Infecciosas, Instituto de Salud Carlos III, 28222 Madrid, Spain; 4Helix BioS. Bioinformatics Solutions, 28049 Madrid, Spain

**Keywords:** DAIR, intracellular, whole genome sequencing, chronic

## Abstract

Background: Persistent and relapsing prosthetic joint infection (PJI) due to *Staphylococcus aureus* presents a clinical challenge. This study aimed to provide an extensive description of phenotypic and genomic changes that could be related to persistence or relapse. Methods: Initial and second *S. aureus* isolates from 6 cases of persistent and relapsing PJI, along with clinical isolates from 8 cases, with favorable outcome were included. All isolates were studied by phenotypic and genotypic approaches. Results: Recurrent *S. aureus* isolates exhibited a significant increase in adhesive capacity, invasion and persistence compared to resolved isolates. No association was found for the presence or absence of certain genes with the persistence or relapse of PJI. All sequential isolates showed identical sequence type (ST). Resistance gene loss during the infection and a great diversity of variants in different virulence genes between the pair of strains, mainly in genes encoding adhesins such as *fnbA*, were observed. Conclusions: *S. aureus*-caused relapse and persistence PJI is associated with bacterial phenotypical and genotypical adaptation. The main paths of adaptation were persistence in the intracellular compartment, and the loss of antibiotic resistance genes and variant acquisition, especially in genes encoding adhesins.

## 1. Introduction 

Prosthetic joint infections (PJIs) are described as difficult to treat [1]. Although the incidence of PJI is not very high, it is still a devastating complication for patients, often requiring surgical intervention and long-term antibiotics, leading to impaired joint function [2]. Debridement, antibiotics and implant retention (DAIR) may be attempted in selected acute cases to minimize the morbidity associated with prosthesis removal and loss of bone stock [1,3]. Unfortunately, however, a large percentage of patients with an acute episode of *Staphylococcus aureus* PJI treated with DAIR eventually develop recurrent or persistent infection [4].

*S. aureus* has been identified as one of the main pathogens responsible for PJI [5]. It is a versatile microorganism with a wide genetic background and virulence factors that play an important role in the pathogenesis of infection and its evolution to persistent or recurrent forms [6,7]. The diversity of virulence factors allows *S. aureus* to adhere to and invade tissue, form biofilm and evade the immune system [7]. The causes of persistent infection are not well established, but possibilities include a limited antibiotic effect, suboptimal surgical management, host immune response or underlying diseases predisposed to recurrence. However, these could also be related to the intrinsic characteristics of the microorganism [8]. In this respect, previous studies have linked certain intrinsic characteristics of *S. aureus* to the persistence or chronicity of the infection, which include biofilm formation in bacterial communities [9,10], persistence in osteoblasts [11], increased adhesion capacity [12], accessory gene regulator (*agr*) dysfunction [13] and the formation of small colony variants (SCVs) [14]. 

Previous studies have focused on analyzing the clinical characteristics of the Staphylococcal–PJI evolution [4,15], but few have delved into the study of the microbiological characteristics and molecular factors that influence PJI development [16,17,18]. We hypothesized that some clinical isolates of *S. aureus* exhibit certain phenotypic and genotypic characteristics that eventually favor persistent or relapsing PJIs. We also investigated potential adaptive mechanisms in recurrent *S. aureus* isolates that allow certain infections to become chronic despite optimal antimicrobial and surgical management. 

## 2. Material and Methods 

### 2.1. Setting and Patients

Our group recently conducted a prospective, multicenter, observational study of 88 cases of PJI caused by *S. aureus* [18]. Most patients were treated with DAIR (debridement < 21 days after symptom onset) and received antibiotic treatment according to the antimicrobial susceptibility profile. Despite this, in six patients, infection persisted or relapsed with the original *S. aureus* strain. We included the initial (recurrent) isolates of the patients with recurrent staphylococcal infection and a second *S. aureus* isolate recovered during antimicrobial treatment in three cases (persistent isolates), and after antimicrobial treatment in another three cases (relapsed isolates). A random selection of clinical isolates from eight patients treated with DAIR who had a favorable clinical course with no evidence of recurrent *S. aureus* infection (resolved isolates) were also included (Table 1).

### 2.2. Antimicrobial Susceptibility Testing

Antimicrobial susceptibility testing was performed using the MicroScan WalkAway System (Siemens, West Sacramento, CA, USA). Minimum inhibitory concentrations (MICs) were interpreted according to EUCAST criteria (Version 7.1, 2017. http://www.eucast.org, access on 10 March 2017). 

### 2.3. Characterization of Hemolytic Activity 

Alpha-hemolysin (*Hla*) activity was analyzed and quantified by a rabbit erythrocyte lysis assay in 96-well plates and by visual inspection and assessment of zones cleared of round bacterial colonies [19]. The activity of the *agr* operon in gamma hemolysin production was measured and categorized as negative, weak, or strong [20]. Beta hemolysin production was also determined [21]. 

### 2.4. Biofilm Formation

Biofilm formation was assessed in triplicate by the 0.7% crystal violet method on microtiter plates using 33% glacial acetic acid as the decolorizing solution [18]. Absorbance was measured at 595 nm. The results were interpreted in accordance with Stepanovic [22].

### 2.5. Fibronectin Adhesion Assay

To measure the adhesion of *S. aureus* to fibronectin, 96-well tissue plates were prepared as described by Seidl et al. [23]. Bound staphylococci were detected by crystal violet staining at 620 nm using an enzyme-linked immunosorbent assay (ELISA) plate reader. 

### 2.6. Cytotoxicity Assay on MG63 Osteoblasts 

As described previously, the cytotoxicity assay was performed on an MG 63 (CRL-1427) human osteoblast cell line [11] using the Cytotoxicity Detection Kit Plus LDH (Roche Applied Science, Indianapolis, IN, USA). The percentage of cell lysis was calculated relative to the value for the maximum LDH release control (cells lysed with a 1% solution of Triton X-100), following the manufacturer’s instructions. 

### 2.7. Adhesion, Invasion and Persistence Intracellular Assay

The MG 63 human osteoblast cell line (CRL-1427) (LGC standard, Manassas, VA, USA) was cultured and maintained in Dulbecco’s modified eagle medium (DMEM) supplemented with 10% fetal calf serum and 100 µg/mL gentamicin [24]. Cells were infected with stationary phase bacterial suspensions at a multiplicity of infection of 100, equivalent to 10^7^ CFU/mL. Inoculum CFU levels were confirmed during each experiment by serial dilution and plating. The first 24-well cell plate was incubated at 37 °C in 5% CO_2_ for 2 h (adhesion). In the next two 24-well culture plates, the medium was replaced with DMEM + 10 µg/mL lysostaphin to lyse all extracellular or adherent staphylococci, and the plates were returned to the incubator for a further 3 h (invasion) and 48 h (persistence). At each of these time points, cells were lysed in 1 mL of sterile water for 10 min; serial dilutions were made; and suspensions were plated to calculate the bacterial CFU. Each isolate was tested in triplicate.

### 2.8. Whole Genome Sequencing

Whole genome sequencing of all *S. aureus* isolates was performed using Nextera (Illumina, San Diego, CA, USA). Libraries were sequenced in a single run on the Illumina NextSeq instrument (150 bp paired-end reads) to generate a coverage of ∼100×. Quality analysis was performed with FastQC (version 0.11.8) (https://www.bioinformatics.babraham.ac.uk/projects/fastqc/, access on 13 June 2018). The resulting reads were assembled with SPAdes (version 3.9) [25] and their quality was evaluated with Quality Assessment Tool for Genome Assemblies (QUAST), version 5.0.2 [26]. Genome annotation was carried out with Prokka (version 1.13.3) [27]. Multilocus sequence typing (MLST) was analyzed by the Center for Genomic Epidemiology (CGE) website (MLST, version 2.0) [28].

Antibiotic resistance genes were obtained by Mykrobe (version 6.1) [29]. The molecular epidemiology analysis was completed (SCCmec and spa type) with SCCmecFinder (version 1.2) and spaTyper (version 1.0) [30]. A pan-genomic analysis of the isolates was performed using Roary (version 3.11.2) [31]. 

Alignment was carried out according to the MLST obtained, using a reference genome belonging to the same or closest CC of the genome under study (Appendix A) [28,29]. Variant calling was performed with Snippy (https://github.com/tseemann/snippy accessed on 18 August 2020) [32] (version 4.4.0), and visualized with IGV (Integrative Genomics Viewer) [33]. The phylogenetic relationship was inferred from maximum likelihood with 1000 replicates, using IQ-TREE (version 1.6.11), after previous deletion of the recombinant regions by Gubbins (version 2.4.1), and was visualized using iTOL (version 5.5) [34,35,36]. Variant analysis was performed with breseq (version 0.33.2) [37]; the variant-calling criteria were adjusted to minimum mapping quality 30, consensus minimum variant coverage 30×, and consensus minimum total coverage 40×. Hypothetical proteins, pseudogenes, intergenic regions and variants in mobile genetic elements (phages) were excluded.

Sequence files were deposited at GenBank under BioProject PRJNA774351 and accession numbers JAJHKM000000000-JAJHNN000000000.

### 2.9. Statistical Analysis

All statistical analysis was performed using GraphPad Prism 8.2.1 software (San Diego, CA, USA). Two-way analysis of variance (ANOVA) was used to compare differences between multiple groups, and the unpaired Mann–Whitney test to compare differences between two samples: *p* values < 0.05 were considered to be significant.

The association between the presence or absence of genes and clinical variables (pan-microbial GWAS) was analyzed with Scoary software (version 1.6.16) [38]. The association between the presence or absence of genes and the clinical evolution of persistent and relapsing PJI was studied by adjusting the *p* value using the Benjamini–Hochberg method (FDR, False Discovery Rate) according to the following criteria: *p* < 0.10 and FDR < 0.05, *p* < 0.05 and FDR < 0.10 and, finally, *p* < 0.05 and FDR < 0.05.

## 3. Results

### 3.1. Genotypic and Phenotypic Comparisons of Isolates from Resolved and Recurrent PJIs 

Isolates leading to recurrent PJI showed increased levels of resistance to all antimicrobials tested (Table 1). One isolate (FailPer118) was methicillin-resistant and harbored SCC*mec* IVc. Three recurrent isolates (FailRel107, FailRel409 and FailPer118) showed resistance to fluoroquinolones, while all isolates in the resolved group were susceptible. FailRel409 isolates were resistant to ciprofloxacin but susceptible to levofloxacin and harbored the *norA* gene, a mutation in the *grlA* gene but not the *gyrA* gene. FailRel107 isolates were resistant to clindamycin but susceptible to erythromycin. Overall, the genotypic and phenotypic antimicrobial resistance correlation was high, with the exceptions of resistance of fosfomycin and cotrimoxazole antibiotics (Table 2). 

We analyzed the intracellular activity of the clinical isolates and found that recurrent *S. aureus* isolates exhibited a significant increase in capacity for adhesion (12.0%, (IQR: 7.8–13.4%) vs. 2.1%, (IQR: 0.9–3.6%), *p* = 0.01), invasion (4.3%, (IQR: 1.9–7.8%) vs. 0.3%, (IQR:0.03–0.5%), *p* = 0.001) and persistence (0.2%, (IQR:0.1–0.9%)) vs. 0.01% (IQR: 0.004–0.002%), *p* = 0.01) compared to those of the resolved PJIs (Figure 1). Hemolytic activity was similar in both groups of isolates (Table 1). On sheep blood agar, all isolates displayed a hemolysis pattern characteristic of *Hla*-induced beta-hemolysis although by rabbit erythrocyte lysis assay, three resolved isolates and one recurrent isolate did not hemolyze the erythrocytes (Appendix A). β-hemolysin with a phage insertion harboring immune evasion genes was detected in all strains except FailRel107, and δ-hemolysin activity was observed in five recurrent isolates (83.3%) and six isolates (75%) recovered from cured patients. Similarly, under conditions of stasis, all *S. aureus* isolates formed biofilm, but it was weak in most and no significant differences between the two groups of isolates was observed (Table 1 and Appendix A). Adhesion to fibronectin in recurrent and resolved isolates was also similar: 0.2 (95% CI: 0.17–0.19) vs. 0.15 (95% CI: 0.13–0.18, *p* = 0.3) (Appendix A) as was cytotoxic activity against MG63 osteoblasts: 49.2% (IQR: 38.9–51.7%) vs. 44%, (IQR: 34.7–51.1%) (Appendix A). 

Figure 2 shows the phylogenetic tree of all isolates causing persistent, relapsing, and resolved PJIs together with the main resistance and virulence genes. The isolates were grouped according to CC and MLST similarity, forming two clearly differentiated clusters. Appendix A shows the virulence genes of all strains causing PJI according to clinical evolution. All isolates harbored genes encoding microbial surface components recognizing adhesive matrix molecules (MSCRAMMs) and genes involved in biofilm formation. The genes *fnbpA* and *fnbpB* were identified in 100 and 85.7% of isolates, respectively, while *cna* was present in only 64.3%. In microbial pan-GWAS, no association was found between the presence or absence of genes and persistent or/relapsing PJI versus resolved PJI. Similarly, no significant variants were identified between strains causing persistent or relapsing PJI and those leading to a favorable outcome.

### 3.2. Phenotypic and Genotypic Changes during Persistent and Relapsing S. aureus PJIs

Sequential isolates from six recurrent PJIs were available for phenotypic and genotypic studies. In the relapsing cases, the second isolate was obtained 20, 55 and 217 days after the end of antimicrobial treatment. All sequential isolates showed an identical sequence type (ST) (Appendix A). All pairs of isolates showed identical antimicrobial resistance patterns except for persistent case 401, which harbored the aminoglycoside resistance gene *aac(6′)-aph(2″)*, implying that FailPer401.1 was resistant to gentamicin, while FailPer401.2 was susceptible. Furthermore, unlike FailPer402.2, FailPer402.1 exhibited the H31N mutation in the *dfrB* gene, which confers resistance to trimethoprim although both strains were susceptible to cotrimoxazole. Finally, unlike FailRel104.2, FailRel104.1 harbored the *blaZ* gene, but we were unable to obtain a definitive resistance phenotype for FailRel104.2 because of the SCV phenotype (Figure 2). 

Analyses of cell adhesion, invasion, and persistence in the sequential isolates revealed no significant differences between initial and second isolates obtained from the same patient (Appendix A). However, we observed certain changes in five pairs of isolates (Figure 3). In four pairs (FailRel104, FailRel107, FailPer401 and FailPer402), the initial isolate showed a greater capacity for adhesion and invasion than the second one (*p* < 0.05), while in just one relapse case (FailRel409), the second isolate showed a greater capacity for adhesion and internalization (adhesion: 1.1 ± 0.8% vs. 6.3 ± 2.8%; *p* < 0.0001 and invasion 1.5 ± 0.2% vs. 5.2 ± 2.7%; *p* = 0.001).

In the same way, all sequential pairs of isolates showed the same *agr* functionality and hemolytic pattern, except for the FailRel104.2 relapse isolate, which showed a loss of *hla* activity and conversion to the SCV phenotype. Surprisingly, we did not observe an increased capacity for biofilm formation in the second isolates (Appendix A), nor did we observe any difference in cytotoxic effect or fibronectin-binding capacity (Appendix A).

With respect to the presence or absence of virulence genes, we found no differences between the initial and second isolates but observed a wide diversity of variants in different virulence genes between strain pairs in relapsing and persistent PJIs (Table 3 and Appendix A). In the FailRel104 and FailRel409 cases, the same non-synonymous SNP was found in the *fnbA* gene in the initial and relapsing strain: Q819H (reference MSSA476 (BX571857.1)). Nevertheless, no differential variants were found between all first and second isolates in the persistent and relapsing cases.

## 4. Discussion

Our study provides an extensive description of the adaptation of *S. aureus* strains during the clinical evolution of persistent and relapsing PJIs and a comprehensive analysis of possible phenotypic and genomics variations of all strains isolated in the same PJI. The availability of sequential isolates offers the opportunity to gain insight into adaptation during *S. aureus* infection. Other authors were limited by a small number of selected cases and so provided only a limited understanding of the general pattern of *S. aureus* evolution during a PJI [17].

A large panel of phenotypic experiments was used to investigate possible pathogenic traits associated with the *S. aureus* isolates that cause recurrent PJI. We found no evidence to support changes in bacterial virulence due to *agr* dysfunction, increased SCV and biofilm formation, adhesion to fibronectin protein or decreased alpha-hemolysin and LDH production. However, we identified one phenotypic feature associated with recurrent but not resolved PJI isolates: an increased capacity for adhesion, invasion and persistence in osteoblasts, which allows *S. aureus* to hide in a protected zone of infection and then re-emerge over time. These results could be important from a clinical point of view since they could indicate that there are isolates with greater potential to cause recurrence, either because infection persists despite treatment or because infection recurs after having been considered cured. After a thorough genetic analysis, we discovered a wide variety of clonal groups and virulence factors in *S. aureus* causing both PJI groups but no genetic molecular signature related to recurrence. The only potential factor that appeared to be associated with PJI recurrence was the higher frequency of quinolone resistance in the recurrent group [39]. 

Next, we investigated possible phenotypic and genotypic traits of sequential isolates recovered from the same patient at different time points of infection. In one case, the loss of antibiotic resistance genes was observed and antibiotic susceptibility increased. This result contrasted with a recent study that showed increased antibiotic resistance in persistent *S. aureus* bacteremia [40]. Interestingly, the FailRel107 strain showed an infrequent pattern of antibiotic susceptibility, with phenotypic susceptibility to erythromycin and phenotypic resistance to clindamycin. This could be caused by the *lnu* gene, which appears to be related to animal clonal lineage CC398 [41]. In addition, this strain did not present the phage carrying the immune system evasion genes, which allows the expression of β-hemolysin [21]. Therefore, it showed turbid halos of hemolysis on blood agar. Phenotypic analysis did not reveal any trait associated with the adaptation of *S. aureus* to persistence or recurrence of infection. In this context, we did not observe any differences between sequential isolates concerning cytotoxicity, biofilm formation or intracellular persistence. This contrasted with a previous study in which recurrent isolates were less cytotoxic, formed more biofilm and persisted longer in the intracellular compartment. However, that study included a small number of strains, which may be problematic for drawing general conclusions [17]. 

With respect to the role of MSCRAMMs in the pathogenesis of *S. aureus* in implant-associated infections [42], in all cases except one (FailPer402), significant variants in adhesin-encoding genes were produced between the initial and second strains although we did not find the same variants in all or most of the strains causing persistent or relapsing cases. Despite the fact that one study on *S. aureus* PJI did not find any association between polymorphisms in fibronectin-binding protein (FnBP) and PJI [43], other studies suggested that antigenic variation in FnBPA could contribute to *S. aureus* evasion of host immune response [44,45]. In connection with this, FailRel104 and FailRel409 showed variants in different MSCRAMMS (*bbp, sdrD, clfA, fnbA* and *fnbB*) and we found the same non-synonymous SNP in the *fnbA* gene between the initial and second strain in both isolates (Q819N). In the intracellular assays, however, the two isolates behaved differently: the initial FailRel104 and the second FailRel409 showed greater adhesion and invasion than the corresponding paired isolate in the sequence. 

A non-synonymous SNP was found in the *fmt* gene of the FailRel104 strain, which has been associated with decreased production of extracellular virulence factors and decreased pathogenesis, which may contribute to increased bacterial survival [46] and persistence. Similarly, a deletion in DNA-3-methyladenine glycosylase, which catalyzes the removal of chemical DNA base lesions, was also observed in FailPer118 cases [47]. This variant could play a role in bacterial adaptation to stressful environments. It is likely that there are reservoirs of latent microorganisms in recurrent infections that are not eliminated by antibiotic treatment or by the host immune system. Under certain conditions, these microorganisms could reactivate and cause recurrence of infection. However, despite the diversity of variants found, we did not observe any in the *agr* operon, the main global regulator linked to the ability to downregulate pro-inflammatory virulence factors and increase the expression of factors promoting persistence. 

The involvement of *S. aureus* SCVs in chronic prosthetic joint infections has been reported [48]. In the course of chronic osteomyelitis, after host cell invasion, *S. aureus* strains have been shown to develop adaptive mechanisms that decrease cytotoxicity and increase the percentage of SCV formation that persist in osteoblasts [16]. In relapsing case 104 (the initial strain was wild phenotype; the second one an SCV), mutations in exotoxins were observed. It has been reported that the SCV phenotype in *S. aureus* is favored by disrupting the electron transport chain with the consequent reduction in ATP production, which inhibits *agr* growth and functionality. Under these conditions, there is a reduction in hemolysin production and an increase in FnBP expression, providing SCV mutants with a high capacity for invasion into new host cells [44]. 

Our study did have some limitations. First, when analyzing the differences between strains, many variants were found in intergenic regions, hypothetical proteins and synonymous SNPs, so we do not know their possible involvement in these infections. Second, many variants were identified between the strains of cases 104 and 409, probably because the reference strain used for analysis (MSSA476) did not belong to the same CC. Third, we did not carry out a transcriptomic analysis of the genome, which may provide more accurate information about the molecular factors of *S. aureus* involved in the persistence and recurrence of PJI. 

In our study, the phenotypic assays showed that the recurrent isolates had greater capacity for adhesion, invasion, and persistence in osteoblasts than those that caused resolved PJI. Conversely, the behavior between pairs of PJI isolates was very heterogeneous, and we were therefore unable to identify any phenotypic pattern associated with persistence or recurrence. 

## 5. Conclusions

Our results showed that *S. aureus* is capable of modifying its genotype during the clinical course of PJI by adapting to recur or persist, especially by the loss of antibiotic resistance genes and the acquisition of variants, mainly in genes encoding adhesins. However, there was no convergence at the genetic level, indicating that adaptation pathways may be episode specific. This, coupled with other key aspects of the pathogenic complexity of PJI, such as the impact of host immunity on the evolution of *S. aureus*, could lead to the progression of infection to a cure or chronicity. 

## Figures and Tables

**Figure 1 antibiotics-11-01119-f001:**
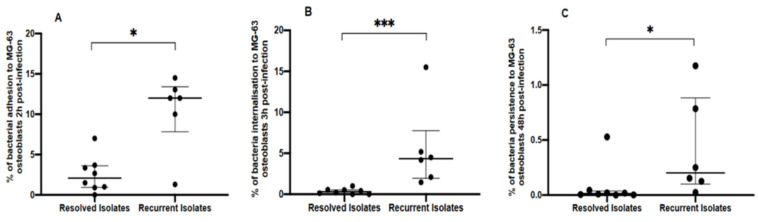
Adhesion (**A**), invasion (**B**) and persistence intracellular assay (**C**) on MG63 osteoblasts of PJI *S. aureus* isolates. All results are expressed as the percentage of the initial strain (≈5 × 10^6^). The horizontal bars denote the median with the interquartile range derived from three independent experiments in triplicate for each clinical isolate. Statistical analysis were performed by using the Mann–Whitney U-test (* *p* < 0.05; *** *p* < 0.001).

**Figure 2 antibiotics-11-01119-f002:**
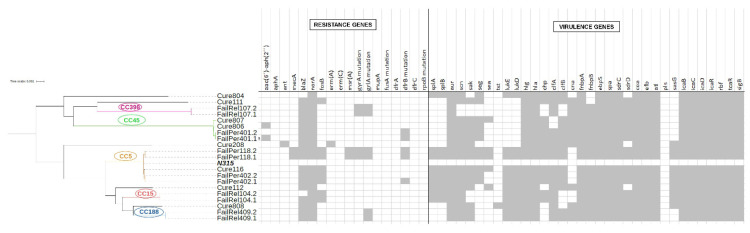
Phylogenetic tree of all the strains causing persistent, relapsing and resolved prosthetic joint infections included in this study, along with the main resistance and virulence genes. Grey boxes indicate the presence of that gene and the white ones its absence.

**Figure 3 antibiotics-11-01119-f003:**
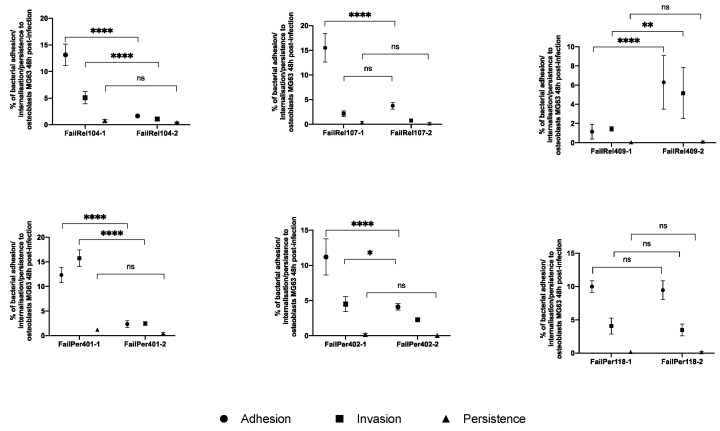
Comparison of the adhesion, invasion and persistence capacities between initial and second isolates of persistent-and relapsing-PJI cases recovered from the same patient. Results represent means ± standard deviation of experiments conducted in triplicate. The statistical analyses were performed by using two-way ANOVA (* *p* < 0.05; ** *p* < 0.01; **** *p* < 0.0001). ns: non-significant (i.e., *p* ≥ 0.05).

**Table 1 antibiotics-11-01119-t001:** Clinical and microbiological characteristics of recurrent (persistence and relapse) and resolved prosthetic joint infection cases included in this study ^1^.

Strain Code	Sex, Age (years)	Infection location	Type of Infection ^2^	Duration of Antibiotherapy (days) ^3^	Outcome ^4^	Antimicrobial Resistance	Hemolytic Activity			Biofilm	ST ^5^
							α-Hemolysin (*hla*)	β-Hemolysin (*hlb*)	δ-Hemolysin (*agr*)		
FailRel104.1	F, 63	Knee	EPI	60	-	PEN	Positive	Negative	Strong	Weak	15
FailRel104.2				-	Relapse	†	Negative	Negative	Strong	Weak	15
FailRel107.1	F, 86	Knee	AHI	66	-	PEN, CL, GEN, CIP, LEV	Positive	Positive	Negative	Weak	398
FailRel107.2				-	Relapse	PEN, CL, GEN CIP, LEV	Positive	Positive	Negative	Weak	398
FailRel409.1	F, 87	Hip	EPI	37	-	PEN, CIP	Positive	Negative	Weak	Weak	188
FailRel409.2				-	Relapse	PEN, CIP	Positive	Negative	Weak	Moderate	188
FailPer401.1	F, 65	Knee	EPI	144	-	PEN, GEN	Positive	Negative	Strong	Weak	45
FailPer401.2				-	Persistence	PEN	Positive	Negative	Strong	Weak	45
FailPer402.1	M, 81	Knee	AHI	78	-	PEN	Positive	Negative	Weak	Weak	5
FailPer402.2				-	Persistence	PEN	Positive	Negative	Weak	Weak	5
FailPer118.1	F, 78	Hip	EPI	25	-	PEN, MET, ER, CIP, LEV	Positive	Negative	Strong	Moderate	125
FailPer118.2				-	Persistence	PEN, MET, ER, CIP, LEV	Positive	Negative	Strong	Moderate	125
Cure116	F, 62	Knee	EPI	79	Resolved	PEN	Positive	Negative	Weak	Weak	5
Cure804	F, 84	Knee	AHI	89	Resolved	‡	Positive	Negative	Weak	Weak	10
Cure806	M, 37	Knee	EPI	81	Resolved	PEN, GEN	Positive	Negative	Strong	Weak	45
Cure111	M, 69	Knee	EPI	82	Resolved	PEN	Positive	Negative	Strong	Weak	30
Cure112	F, 83	Knee	AHI	90	Resolved	PEN	Positive	Negative	Negative	Weak	6
Cure208	M, 53	Hip	EPI	93	Resolved	PEN, ER, CL	Positive	Negative	Negative	Strong	509
Cure807	F, 63	Knee	EPI	57	Resolved	PEN	Positive	Negative	Strong	Weak	45
Cure808	F, 72	Knee	EPI	100	Resolved	PEN	Positive	Negative	Strong	Weak	852

^1^ All patients had an acute PJI and were managed with the DAIR strategy (debridement <21 days after onset of symptoms). ^2^ Type of infection; EPI: Early Postoperative Infections; AHI: Acute Haematogenous Infections. ^3^ The duration of antimicrobial therapy among cases of persisting infection was considered from debridement until antimicrobial withdrawal or until salvage surgical treatment. ^4^ Persistence: the second *S. aureus* strain was re-isolated in the course of antibiotic treatment recovered from the same patient. Relapse: the second *S. aureus* strain was re-isolated after the end of the antibiotic treatment. For strains (FailRel104.2, FailRel107.2 and FailRel409.2) the time to relapse was 217, 55 and 20 days, respectively. Abbreviations: AHI, Acute Haematogenous Infections; CIP, ciprofloxacin; CL, clindamycin; EPI, Early Postoperative Infection; ER, erythromycin; F, female; GEN, gentamicin; LEV, levofloxacin; M, male; MET, methicillin; PEN, penicillin; ^5^ ST, sequence type. † We could not obtain definitive phenotype resistance for FailRel104.2 because of its small colony variants (SCV) phenotype. ‡ The Cure804 isolate was susceptible to all antibiotics tested.

**Table 2 antibiotics-11-01119-t002:** Correlation of phenotypic and genotypic antimicrobial resistance of 14 PJI isolates ^a^.

Antibiotic Type	Phenotypic Resistance N (%)		Genotypic Resistance N (%)	
Betalactams	Penicillin Methicillin	13 (92.9) 1 (7.1)	*blaZ* *mecA*	13 (92.9) 1 (7.1)
Aminoglycosides	Gentamicin	3 (21.4)	*aac(6′)-aph(2″)* *aphA* *ant*	2 (14.3) 0 (0.0) 1 (7.1)
Fluoroquinolones	Ciprofloxacin Levofloxacin	3 (21.4) 2 (14.3)	*norA* *gryA* mutation (S8AL) *grlA* mutation (S80F)	14 (100) 2 (14.3) 3 (21.4)
Macrolides, lincosamides, streptogramin B	Erythromycin Clindamycin	2 (14.3)2 (14.3)	*erm(A)* *erm(C)* *msr(A)*	1 (7.1) 0 (0.0) 1 (7.1)
Trimethoprim	Cotrimoxazole	0 (0.0)	*dfrA* *dfrB* mutation (2 H31N, 1 F99Y) *dfrC*	0 (0.0) 3 (21.4) 0 (0.0)
Fosfomycin	Fosfomycin	0 (0.0)	*fosB*	5 (35.7)
Mupirocin	Mupirocin	0 (0.0)	*mupA*	0 (0.0)
Fusidic acid	Fusidic acid	0 (0.0)	*fusA* mutation	0 (0.0)
Rifampin	Rifampin	0 (0.0)	*rpoB* mutation	0 (0.0)

^a^ In relapsing or persistent cases, it has only been included the first strain for this analysis.

**Table 3 antibiotics-11-01119-t003:** Variants between relapsing or persistent isolates from the same patient.

Isolates	Nucleotide Position	Initial Isolate	Relapsing/ Persistent Isolate	Annotation	Type of Mutation	Description (Gene)
FailRel104	82110	A	C	G57G	SYN	lipoprotein
	126756		Δ8 bp	83–90/741 nt	DEL	GntR family regulatory protein
	241537–243250				4 MBS, 3 NSYN, 5 SYN	staphylocoagulase precursor
	428158–431375				6 MBS, 11 NSYN, 4 SYN	exotoxin
	437491–437497				3 NSYN	exotoxin5
	442703, 442706				SYN, NSYN	exotoxin
	446421–448928				5 NSYN, 6 SYN, 1 MBS	lipoprotein
	479985, 479991				2 SYN	glutamate synthase, small subunit
	560689		Δ3 bp	343–345/1401 nt	DEL	cysteinyl-tRNA synthetase
	598737	T	A	A364A	SYN	serine-aspartate repeat-containing protein D (*sdrD*)
	604977	C	T	D947D	SYN	bone sialoprotein-binding protein
	842802, 842812				MBS, SYN	clumping factor A (*clfA*)
	946908–947417				3 SYN, 1 DEL, 1 MBS	transport system extracellular binding lipoprotein
	1059255		+TGTTGGTTTCGACGGTGT	1279/3753 nt	INS	bifunctional autolysin precursor
	1388164, 1388182				MBS, SYN	topoisomerase IV subunit A
	1883013	C	T	A234T	NSYN	serine protease
	2088541–2088547				3SYN	autoinducer sensor protein
	2275393–2275829				2 DEL, 4 NSYN, 1 SYN, 1 MBS	hyaluronate lyase precursor 2
	2464027		Δ4 bp	126–129/780 nt	DEL	extracellular solute-binding lipoprotein
	2538425–2538596				6 SYN, 1 NSYN, 4MBS	lipoprotein
	2557255–2558646				2 SYN, 2MBS	fibronectin-binding protein B precursor (*fnbB*)
	2560605–2562933				5 NSYN, 7 SYN, 2 MBS, 2 DEL	fibronectin-binding protein A precursor (*fnbA*)
FailRel107	915183	A	G	G139D	NSYN	lpxtg-motif cell wall anchor domain
	1642042	T	C	Y18Y	SYN	integrase
	1699959	A	C	T582T	SYN	penicillin-binding Protein dimerisation domain family
FailRel409	81598–83027				6 NSYN, 6 SYN, 3 MBS	lipoprotein
	98820	G	A	G298G	SYN	immunoglobulin G binding protein A precursor
	243016	C	T	N532N	SYN	staphylocoagulase precursor
	431559, 431562				2 NSYN	exotoxin
	598737	A	T	A364A	SYN	serine-aspartate repeat-containing protein D (*sdrD*)
	843649	G	T	V867V	SYN	clumping factor A (*clfA*)
	1059255	+TGTTGGTTTCGACGGTGT		1279/3753 nt	INS	bifunctional autolysin precursor
	1148233		+68 bp	931/939 nt	INS	ribonuclease HIII
	1622493, 1622511				DEL, NSYN	DNA primase
	2088515		3 bp→CTC	610–612/1293 nt	MBS	autoinducer sensor protein
	2496948–2497602				7 SYN, 1 NSYN, 7MBS	protein flp
	2538455, 2539355				SYN, MBS, NSYN	exported protein (lipoprotein)
	2560599–2562279				5 SYN, 3 NSYN	fibronectin-binding protein A precursor (*fnbA*)
	2567893–2567901				2 SYN, 2 NSYN	MerR family regulatory protein
	2700008, 2700011				2 SYN	fibrinogen and keratin-10 binding surface anchored protein
FailPer401	19218		+G	874/1968 nt	INS	DHHA1 domain protein
	103426	T	A	I503F	NSYN	N-acetyl-ornithine/N-acetyl-lysine deacetylase family protein
	199157		Δ131 bp	730-860/960 nt	DEL	cation diffusion facilitator transporter family protein
	1197044	A	C	K309Q	NSYN	methionyl-tRNA formyltransferase (*fmt*)
	1598596		Δ8 bp	695-702/1020 nt	DEL	maltose operon transcriptional repressor-like protein
	2042765		Δ107 bp	1167-1273/1539 nt	DEL	sodium/proline symporter (*putP*)
	2215085	A	T	N169I	NSYN	ATP synthase F1, gamma subunit (*atpG*)
	2397306		+T	193/348 nt	INS	transcriptional regulator family protein
FailPer402	No mutations were found in the strains belonging to the persistent PJI 402.					
FailPer118	1696986	(AATG)3→4		11/561 nt	DEL	DNA-3-methyladenine glycosidase

Abbreviations: bp, base pair; DEL, deletion; INS, insertion; MBS, multiple-base substitution; NSYN, non-synonymous SNP; nt, nucleotide; SYN, synonymous SNP. Variants in a single protein were grouped and, therefore, no annotation was shown in those variants.

## Data Availability

The data presented in this study are available in Appendix A.

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
