# Peer review of "Microbiological and Molecular Features Associated with Persistent and Relapsing Staphylococcus aureus Prosthetic Joint Infection"

_antibiotics, 2022, doi:10.3390/antibiotics11081119_

Round 1

Reviewer 1 Report

The authors conducted a prospective, multicenter, observational study of 88 cases of prosthetic joint infections (PJIs) caused by Staphylococcus aureus (S. aureus). They suggested that recurrent S. aureus isolates exhibited a significant increase in the capacity of adhesion, invasion, and persistence compared to resolved isolates. They also found that a resistance genes loss during the infection and a great diversity of variants in different virulence genes between the pair of strains, mainly in genes encoding adhesins, such as fnbA were observed. The findings help for a better understanding of the bacterial phenotypical and genotypical adaptation in PJIs. Overall, the writing is good. I only have a few minor comments.

MINOR Comments

There are two “..” in the title of Table 1.

Why are the clinical characteristics of the first patient bolded?

In the legends of Table 1, there should have other spaces after 2 and 4.

Figure 3 should show the non-significant (ns) line in the figure.

Reviewer 2 Report

The manuscript studied the phenotypic and genomic changes that are related to persistent and relapsing prosthetic joint infection duo to Staphylococcus aureus.  To aim it, the authors collected samples from relapsing/persistent PJI cases and cure cases. The results showed that S. aureus relapse and persistence PJI is associated with a bacterial phenotypical and genotypical adaptation.

There are still some minor comments.

1.There is grammar problem in line27-line28.

2. Line 156 showed that “three recurrent isolates showed resistance to fluoroquinolones…”.  To be clear, strain code should be added, e.g. FailRel104.

3. Table2 is the correlation between genotypic and phenotypic antimicrobial resistance, as described in line160-line161. What is the information and conclusion from Table2. The authors need to describe more.

4. Line235-line239 described Figure 3. Here only showed four isolates… one isolate…..without the case number. I would suggest to either add case number or add A/B/C to figures.
